# Platelet Lysate Induces in Human Osteoblasts Resumption of Cell Proliferation and Activation of Pathways Relevant for Revascularization and Regeneration of Damaged Bone

**DOI:** 10.3390/ijms21145123

**Published:** 2020-07-20

**Authors:** Van Thi Nguyen, Marta Nardini, Alessandra Ruggiu, Ranieri Cancedda, Fiorella Descalzi, Maddalena Mastrogiacomo

**Affiliations:** 1Department of Experimental Medicine (DIMES), University of Genova, 16132 Genova, Italy; nguyenthivan.sha49@gmail.com (V.T.N.); alessandra.ruggiu1982@gmail.com (A.R.); descalzi.fiorella@gmail.com (F.D.); 2Department of Internal Medicine (DIMI), University of Genova, 16132 Genova, Italy; nardinimarta88@gmail.com; 3Biotherapy Unit, Ospedale Policlinico San Martino, 16132 Genova, Italy; 4Endolife S.r.l. 16132 Genova, Italy; ranieri.cancedda@unige.it; 5Center for Biomedical Research (CEBR), University of Genova, 16132 Genova, Italy

**Keywords:** platelet lysate (PL), platelet factors, angiogenesis, osteogenesis, HIF-1α, STAT3, VEGF

## Abstract

To understand the regenerative effect of platelet-released molecules in bone repair one should investigate the cascade of events involving the resident osteoblast population during the reconstructive process. Here the in vitro response of human osteoblasts to a platelet lysate (PL) stimulus is reported. Quiescent or very slow dividing osteoblasts showed a burst of proliferation after PL stimulation and returned to a none or very slow dividing condition when the PL was removed. PL stimulated osteoblasts maintained a differentiation capability in vitro and in vivo when tested in absence of PL. Since angiogenesis plays a crucial role in the bone healing process, we investigated in PL stimulated osteoblasts the activation of hypoxia-inducible factor 1-alpha (HIF-1α) and signal transducer and activator of transcription 3 (STAT3) pathways, involved in both angiogenesis and bone regeneration. We observed phosphorylation of STAT3 and a strong induction, nuclear translocation and DNA binding of HIF-1α. In agreement with the induction of HIF-1α an enhanced secretion of vascular endothelial growth factor (VEGF) occurred. The double effect of the PL on quiescent osteoblasts, i.e., resumption of proliferation and activation of pathways promoting both angiogenesis and bone formation, provides a rationale to the application of PL as therapeutic agent in post-traumatic bone repair.

## 1. Introduction

Regenerative Medicine, which is the reactivation of regeneration pathways to restore tissue structure and function, is the new frontier for the treatment of tissues damaged by trauma or pathologies. After a wound, in almost all tissues, the first events are the blood extravasation with the formation of a hematoma and the onset of a local inflammatory response. The new microenvironment triggers a cascade of events eventually leading to healing of the damaged tissue. During clot formation, plasma turns to serum and platelet activation and degranulation occur, resulting in the release of factors and cytokines promoting cascade events, including resumption of proliferation of some resident cells [1,2]. Given the cell proliferation induction exerted by the platelet content, platelet derived products such as, platelet rich plasma (PRP) and platelet lysate (PL) obtained by lysis of platelets concentrated in a small volume of plasma, were produced and proposed as fetal calf serum (FCS) replacement, in different cell culture systems.

Also considering the in vitro evidence, platelet derivatives were directly adopted in the clinical practice for the treatment of different tissue defects. In spite of existing discrepancies in the literature regarding the use of platelet derived products [3], encouraging results were reported by the PRP use in chronic skin wounds [4,5,6,7], in dental and maxillofacial surgery [8,9], in orthopedics [10,11,12], and in ophthalmology [13,14]. Beneficial effects were also shown in Sport Medicine [12,15] and in clinical trials for the treatment of cartilage degenerative diseases performed with either autologous PRP [16] or autologous PL [17].

We reported that the platelet factors promoted the activation of the cell proliferation machinery and the cell-cycle re-entry of several types of cells, including chondrocytes and chondroprogenitors [18,19,20], human umbilical vein endothelial cells [21,22], bone marrow and umbilical cord mesenchymal stromal cells (MSCs) [21,23], adipose tissue cells [21,23,24], and amniotic fluid cells [25].

Given the particular interest of our group in treatment of bone and cartilage defects, in the attempt to better understand the putative regenerative effect of platelet-released molecules in bone healing and to investigate the cellular and biochemical cascades of events involving the resident osteoblast population, in a previous paper, we focused on the in vitro response of human primary osteoblasts to the platelet factors [26]. In a wound, the plasma turns to serum and platelet activation and degranulation occur in the presence of serum. In the same article we reported that the addition of PL made in physiological buffer to a culture medium, containing FCS, promoted the migration of a large number of actively proliferating cells out of cultured bone chips. Moreover, the supplement of PL to the medium of quiescent growth arrested osteoblasts induced a cell morphology change and the resumption of cell proliferation together with a transient increase of the inflammatory response—nuclear factor kappa-light-chain-enhancer of activated B cells (NF-κB) activation and cyclooxygenase (COX-2) induction—and the secretion of prostaglandin E_2_ (PGE_2_) and pro-inflammatory cytokines. The resumption of cell proliferation correlated with extracellular signal-regulated kinases (ERKs) and protein kinase B (Akt) up-regulation, induction of cyclin D1 and phosphorylation of retinoblastoma (Rb).

At the time of a bone fracture or trauma, bone cells, including progenitors and steady state osteoblasts, are exposed to a burst of platelet-derived factors for a limited period of time. In the present study, none or very slow dividing “aged” osteoblasts derived from bone chips of elderly patients and maintained in the presence of FCS for several culture passages, were supplemented with PL before being reverted to an only FCS regimen. We observed a resumption of proliferation after PL stimulation and a return to a none or very slow dividing condition when the PL was removed. We studied the differentiation capability in vitro and in vivo of the FBS + PL cultured osteoblasts when they were tested in the absence of PL. Then, considering that the bone healing is a complex event, also involving angiogenesis recovery [27], we investigated in the PL stimulated osteoblasts the activation of signal transducer and activator of transcription 3 (STAT3) and hypoxia-inducible factor 1 (HIF-1) pathways and the vascular endothelial growth factor (VEGF) secretion, relevant in both bone regeneration and angiogenesis [28,29].

Concluding, we studied the effect of platelet released molecules on cultured osteoblasts to mimic the physiological response of bone cells when a wound occurs and the cells are stimulated by a burst of platelet factors in term of their resumption of proliferation retaining differentiation capability. Moreover, we investigated the osteoblast involvement in the activation of pathways promoting angiogenesis and osteogenesis, eventually leading to bone healing.

## 2. Results

### 2.1. PL Induces a Strong Proliferative Response in Aged Slow Proliferating Cultured Osteoblast

In a previous study, we observed that a PL treatment of low passage cultured osteoblasts was boosting cell proliferation without affecting the osteogenic differentiation potential [26]. To mimic the physiology of the bone healing after a wound, when bone osteoblast exposure to the platelet release is limited in time, in the present study, cultures of slowly or not proliferating primary osteoblasts maintained in standard medium in the presence of FCS for 5 culture passages (that is for about 4–5 weeks and 2–3 cell doublings) were split in two. One half was maintained in medium supplemented with FCS, while the other half was cultured in medium supplemented with FCS + PL. When the cells reached sub-confluence, they were detached, counted and re-plated at a concentration 1 × 10^4^ cells/cm^2^. Both cultures were continued for 2 weeks. At that time the FCS + PL culture was split in two: one half was maintained in the same medium while the other half was transferred in medium supplemented with only FCS. All three cultures were continued for additional 2 weeks. For all culture conditions, cells were plated in several Petri dishes and the number of cells was determined at different time intervals by counting the cells present in two of these dishes. Results are expressed as the average of three independent experiments performed on different single-donor primary cultures performed in duplicate ± standard deviation (SD) value (Figure 1A). When cells cultured in FCS were switched to FCS + PL they showed a strong increase of their proliferation rate (about one doubling/2 days). However, when the FCS + PL supplemented cells were re-transferred to FCS only supplemented medium they stopped to proliferate and assumed a proliferation behavior comparable to the one of the cells maintained throughout the all culture in the only FCS supplemented medium (less than one doubling/2 weeks). As already observed in our previous publication [26], cells in PL supplemented medium (FCS + PL) presented a smaller and more elongated morphology than cells maintained in, or reverted to, a standard medium (FCS) (Figure 1B).

### 2.2. PL Stimulated Osteoblast Maintain Differentiation Potential

Proliferation and differentiation are usually considered two alternative options for the cells. Therefore, the induction of cell proliferation by PL posed an issue about the differentiation ability of the PL stimulated “aged” osteoblasts after being deprived of the PL. In our previous study, we observed that the presence of PL in the medium of the growth arrested osteoblasts during the 21 days osteogenic differentiation assay did not affect the osteoblast differentiation. In the present study, we tested the osteogenic differentiation potential of cultures of osteoblasts expanded in the presence of FCS, proliferation induced by PL for 2 weeks and reverted to only FCS condition or expanded in the presence of FCS and maintained in the presence of FCS (control). As shown by both the in vitro (Figure 2) and the in vivo (Figure 3) assays, an osteogenic differentiation was observed for both types of cultures. However, in the in vitro assay, deposition of calcium mineral was observed in the PL stimulated osteoblasts earlier than in osteoblast continuously maintained in only FCS supplemented medium (Figure 2A,B).

### 2.3. PL Induces the Stabilization of Hypoxia-Inducible Factor 1-Alpha (HIF-1α) and the Activation of Signal Transducer and Activator of Transcription 3 (STAT3)

In a tissue wound, the vascular injury leads to a stop of the blood flow and to the consequent ischemia and hypoxia. Hypoxia induces the stabilization of HIF-1α, a transcription factor that accumulates in the wounded tissue cells, relocates to nucleus and combines with HIF-1β to form an active HIF-1 complex binding to hypoxia-response element (HRE) sequences of target genes including VEGF [30]. Hypoxia-inducible factor 1-alpha stabilization can be induced also in normoxic conditions by some cytokines, growth factors, and microbe-derived components [31]. Indeed, the HIF-1 complex is able to induce the expression of genes necessary for cell survival and metabolism under a variety of hostile conditions [32].

We here report that, in subconfluent cultures of osteoblasts maintained in normoxic conditions, PL induced a significant increase in the level HIF-1α already after 4 h exposure and that the level of HIF-1α progressively decreased after 8 and 24 h (Figure 4 and Appendix A). A similar timing was observed also for the appearance of the phosphorylated STAT3, another transcription factor involved in bone cells differentiation and fracture healing [33] while, as previously reported the expression of the cyclin D1, here tested as control induced protein, reached the highest level after 8 h.

### 2.4. Functional Expression of HIF-1α and STAT3

The observed stabilization of HIF-1α and activation of STAT3 were functional. HIF-1α and the phosphorylated form of STAT3 were translocated to the nucleus, leading to a highest amount of nuclear proteins 24 h after the PL treatment (Figure 5A,B and Appendix A). We also investigated HIF-1α binding to nuclear DNA and we showed that at 4 h there was already an increase of HIF-1α binding to the DNA responsive specific sequences, although the highest level of binding was observed after 24 h (Figure 5C).

### 2.5. The HIF-1α Inhibitor Acriflavine Inhibits Osteoblast Proliferation

Recent studies revealed a role played by hypoxia and HIF pathway for the re-entry of cells in the cell cycle and their contribution to organ regeneration [34,35,36,37,38,39]. We wanted to investigate a possible involvement of HIF-1α in the PL induced osteoblast proliferation. Therefore sub-confluent “aged” osteoblasts were treated with PL in the absence and in the presence of the HIF inhibitor acriflavine, a direct inhibitor of HIF-1 that prevents HIF-1 dimerization [40]. The effect of the inhibitor on the PL induced resumption of proliferation by osteoblasts was determined. The presence of acriflavine partly inhibited the PL induced osteoblast proliferation in agreement with a possible control by HIF-1 also of the re-entry in cycle of osteoblasts (Figure 6).

### 2.6. PL Induces Synthesis and Secretion of VEGF

Hypoxia-inducible factor 1α, greatly up-regulated under low oxygen tension and it is known to promote transcription of various factors, including VEGF considered the angiogenic growth factor par excellence, regulating vasculature growth [27]. Interestingly, angiogenesis and osteogenesis being coupled events, VEGF was also described to play a significant role in the control of osteoblast differentiation and bone formation during bone repair [29]. Given that HIF-1α was induced by PL in a normoxic condition, we wanted to compare secretion of VEGF in osteoblasts not stimulated and stimulated by a 24 h treatment with PL. By Western blot analysis we observed the induction of VEGF specific bands that appeared already 4 h after PL exposure and reached the highest level of expression at 8–24 h (Figure 7A and Appendix A). In agreement with this observation, the secretion of VEGF was more than doubled in the osteoblasts treated with PL compared to control osteoblasts maintained in FCS (Figure 7B).

## 3. Discussion

Given the progressively increasing adoption of the Platelet Rich Plasma (PRP), i.e., a concentration of platelets in plasma, in orthopedics, our goal was to investigate the stimulatory activity of the platelet component on osteoblasts, focusing on the activity of a pure platelet lysate fraction.

The microenvironment changes taking place after a wound comprise a transition from plasma to serum and back to plasma. In a previous study, we showed that, when to osteoblasts cultured in medium containing calf serum was added a pure PL fraction, the cells underwent a transient inflammatory response together with an increased production and secretion of PGE_2_ and inflammatory cytokines [26]. Moreover, with the addition of PL, growth arrested osteoblasts resumed proliferation maintaining their differentiation potential. In agreement with this finding, a much larger number of osteoprogenitors outgrew from cultured bone chips when the cultures were performed in the presence of FCS + PL compared to cultures performed in the presence of only FCS [26]. In the present study, we extended our previous observations considering the physiology of the bone healing where osteoblasts are in contact with PL for a limited time. We characterized the cells after exposure to PL in terms of proliferation and differentiation and we investigated, in the same cells, the capacity of PL to activate pathways and to induce the expression of factors involved in revascularization and regeneration of the bone.

### 3.1. PL Induced Proliferation of Quiescent Osteoblasts

In vivo, osteoblasts are entrapped in their own bone matrix and do not proliferate. However, following a fracture or a serious damage, steady state osteoblasts (precursors or more mature cells) are exposed to the platelet released factors, resume proliferation and participate in the cascade of events leading to the fracture repair. To mimic the fracture environment and considering that the exposure of the damaged tissues to the platelet-derived factors is limited in time, we expanded osteoblasts in media containing FCS until they were none or very slow proliferating (“aged” osteoblasts). When PL was added to the culture medium already containing FCS, cell proliferation rate was highly increased, but, when the PL was removed from the culture medium, osteoblasts resumed a none or a very low proliferating rate and behaved like control osteoblasts maintained for the whole time in FCS only. However, when properly induced, the PL stimulated and reverted to only FCS regimen osteoblasts maintained the ability to differentiate in vitro and in vivo in a permissive environment. This finding raises the possibility that, at the time of a fracture or a bone damage, resident cells, possibly precursors and growth arrested osteoblasts, after having been exposed to the PL burst, resume proliferation and expand, except to return to a differentiated stage at the end of the PL stimulus, thus, concurring to the new bone deposition.

### 3.2. PL Induced in Osteoblasts Stabilization, Nuclear Translocation and DNA Binding of HIF-1α, and VEGF Secretion

Tissue injury leads to the damage of blood vessels and the disruption of local blood perfusion resulting in a hypoxic state that requires the adaptive response of tissue cells to the modified environment. Hypoxia-inducible transcription factors (HIFs) are a complex of factors that regulate cell response to hypoxia. In particular, the HIF-1α subunit, present in the cell cytoplasm and degraded by the ubiquitin system, is stabilized in the hypoxic condition and migrates to the nucleus to join with HIF-1β and co-activators thus forming a complex, that activates the transcription of a multitude of genes important for cell survival and metabolism [41]. Although the activation of the HIFs is generally considered the result of a hypoxia, the formation of the HIF complex is also induced in normoxic conditions by several stimuli occurring in injury and hostile environments, such as specific cytokines and growth factors, reactive oxygen and nitrogen species and microbial components [31].

In a recently published paper, we reported that PL induced an increase of the HIF-1α protein, its nuclear relocation and its binding to specific DNA responsive elements in cartilage cells cultured in normoxic conditions [19]. Here, we report that in normoxic conditions, PL induced an increased expression of HIF-1α, its translocation to the nucleus and its binding to specific DNA responsive elements also in osteoblasts. Likewise, in the cultured osteoblasts, we observed activation and nuclear translocation of STAT3 and an increased secretion of VEGF. These findings suggest that PL stimulated osteoblasts could participate to the bone healing by turning on pathways and factors known to induce endothelial cell activation leading to angiogenesis, but also directly involved in the osteogenesis process.

It has been reported that HIF-1α enhanced bone formation by regulating osteoblast differentiation [42]. As a matter of fact, a marked decrease in bone volume was observed in mice lacking HIF-1α in osteoblasts [43]. The occurrence of a decreased bone mass in mice lacking HIF-1α in osteoblasts was confirmed by Shomento et al. [42] and they suggested that HIF-1α is critical for coupling angiogenesis to osteogenesis during endochondral ossification. Other authors reported that HIF-1α is a pro-osteogenic factor for woven bone formation after damaging loading [44]. A role of HIF-1α in the control of osteoblast–osteoclast crosstalk was also reported [45].

Undoubtedly, HIF-1α pathway is a key component of skeletal development and activation of the HIF complex is critical for coupling angiogenesis to osteogenesis during bone formation and for maintaining bone homeostasis [46]. In transgenic mice, the overexpression of HIF-1 in mature osteoblasts, strongly increased, via VEGF, angiogenesis and osteogenesis, two processes tightly coupled in bone healing. Indeed, VEGF is playing a significant role both in osteoblasts differentiation and in the angiogenesis induction [29,47,48]. Moreover, in a recent publication, we reported that PL has a direct effect on endothelial cells by inducing their proliferation while maintaining their differentiation potential [22].

### 3.3. PL Induced Activation of STAT3 and Proliferation in the Osteoblasts

Signal transducer and activator of transcription 3 is a ubiquitously present transcription factor. Upon activation, STAT3 is phosphorylated on tyrosine residue and forms homo- or heterodimers that translocate to the nucleus, where they induce the transcription of genes that mediates cell proliferation, differentiation, survival, apoptosis, and cellular immunity [49,50,51]. Among several biological functions, STAT3 plays an important role in the regulation of angiogenesis at the transcriptional level, including the transcriptional activation of VEGF, both in physiological and pathological conditions. In MSC, the peak of VEGF expression occurs after 3 days of hypoxia. Hypoxia also enhanced expression and phosphorylation of STAT3 and inhibiting STAT3 reversed the VEGF induction [52]. Signal transducer and activator of transcription 3 signaling is important and necessary for endothelial cell proliferation, migration, and microvascular tube formation [33].

Signal transducer and activator of transcription 3 signaling plays a crucial role also in bone formation and homeostasis. In humans, STAT3 mutations reduce bone mass and increase incidence of minimal trauma fractures [53,54]. In transgenic mice, specific inactivation of STAT3 in osteoblasts and osteocytes decreased mechanical load-driven bone formation [55]. CoCl_2_, a mimic of hypoxia, enhanced migration in MSCs and promoted osteogenic differentiation of MSCs and bone defect healing via STAT3 phosphorylation upregulation [28,52]. An increase in STAT3 phosphorylation was also observed during differentiation of human osteoprogenitors. Moreover, the upregulation of JAK2, an upstream of STAT3 signaling, stimulated osteogenic differentiation of progenitor cells and bone defect healing [56].

Our finding that PL induced in osteoblasts activation and nuclear translocation of STAT3 in normoxic conditions is in line with the literature data and strengthen the concept of using PL to enhance bone healing.

Lastly, it is noteworthy that the activation and the nuclear translocation of HIF-1α seems relevant also for the cell cycle activation and the resumption of proliferation in different types of cells [34,35,37,39,57]. In particular, it was reported that HIF-1α can control osteoblast proliferation [42]. The HIF-1 inhibitor acriflavine was reported to inhibit tumor growth [40]. We showed that, in quiescent chondrocytes, the cell proliferation induced by PL was inhibited by the acriflavine treatment that blocked the HIF-1 binding to the HIF-1 responsive elements [19]. Here we described that acriflavine also partly inhibited the proliferation of the PL stimulated osteoblasts.

## 4. Materials and Methods

### 4.1. Materials

Iscove medium (Iscove MEM), L-glutamine, trypsin/ethylenediaminetetraacetic acid (EDTA), and penicillin/streptomycin were obtained from EuroClone S.p.A. (Milan, Italy). Fetal Calf Serum (FCS), collagenase I, and collagenase II were purchased from ThermoFisher Scientific (Waltham, MA, USA). Ascorbic acid, dexamethasone, β-glycerophosphate, tyazolyl blue (MTT), Alizarin Red S was purchased from Sigma Aldrich (St. Louis, MO, USA). Dispase II was purchased from Roche (Basel, Switzerland).

Antibodies α-Cyclin D1 and α-Actin were from Santa Cruz Biotechnology (Dallas, TX, USA), HIF-1α from BD Bioscience (San Jose, CA, USA), phospho-STAT3 and STAT3 from Cell Signaling Technology (Danvers, MA, USA), horseradish peroxidase (HRP) conjugated anti mouse and anti-rabbit antibodies and enhanced chemiluminescence (ECL) were purchased from GE Healthcare (Chicago, IL, USA), VEGF polyclonal antibody from ThermoFisher Scientific (Waltham, MA, USA). Peroxidase conjugated anti goat antibody (1:10,000) from Jackson Immunoresearch (Cambridgeshire, United Kingdom), TransAM HIF-1 kit was from Active Motif (Carlsbad, San Diego, CA USA), Human VEGF Quantikine ELISA kit was provided by R&D System (Abingdon, UK).

### 4.2. Preparation of Platelet Lysate Preparation

Platelet Lysate (PL) was prepared according to [23] from healthy blood wastes of Blood Transfusion Center of IRCCS Ospedale Policlinico San Martino (Genova, Italy). At the time of blood donation, all donors provided the written informed consent for use of donated blood for clinical and scientific applications. PL was produced from pools of buffy coats derived from several blood donations (at least 25 donations) in order to minimize variations among donors. Buffy coat pools were centrifuged at low speed and the lower phase, enriched in platelets (PRP), was separated and centrifuged at higher speed to sediment platelets. Platelet pellets were washed 3 times with physiological saline (0.9% *w/v* NaCl) to eliminate plasma-derived contaminants before being re-suspended in PBS at a concentration of 10 × 10^6^ platelets/µL. The resulting platelet suspensions were subjected to 3 consecutive freeze-thaw cycles to activate and lyse platelets. Broken platelet membranes and debris were removed by a high-speed centrifugation and supernatants, containing the cocktail of factors released by the platelets, were collected and stored in aliquots at −80 °C until use. Being plasma and serum free, the PL did not require addition of heparin to the cell culture medium at the time of its use.

### 4.3. Osteoblast Isolation

Samples of trabecular bone were removed from the femoral head of adult patients who underwent hip joint replacement surgery at the San Martino Hospital Orthopedic Clinic (Genoa, Italy). The informed consensus of the patients and the approval of the institutional ethics committee were preliminary obtained. The patient age ranged from 41 to 85 years (Appendix A). To isolate primary osteoblasts, trabecular bone samples were cleaned of adherent soft tissues and cut into small fragments (2 mm × 2 mm). The fragments were washed in PBS and left overnight in Ringer solution (147 mM NaCl, 4 mM KCl, and 1.13 mM CaCl_2_ in water). Subsequently, fragments were first digested in 1 mg/mL trypsin/Ringer solution for 10 min at 37 °C, then in 2 mg/mL dispase/Ringer solution for 20 min at 37 °C and finally in 3 mg/mL collagenase I/Ringer solution twice for 30 min at 37 °C. Cells released by the collagenase digestion were washed twice in PBS, plated and grown in Iscove medium supplemented with 2 mM L-glutamine, 50 mg/mL penicillin/streptomycin and 10% FCS (complete medium). The medium was changed 24 h after the procedure and then every 2–3 days. Subconfluent cells were trypsin detached, diluted with fresh medium and re-plated. Cultures were performed in a controlled environment at 37 °C in a humidify atmosphere of 5% CO_2_.

### 4.4. Proliferation Assay

#### 4.4.1. Crystal Violet Staining

Staining was performed as described by T. Van Nguyen et al. [19]. Briefly, cells were seeded in 96 multi-well plates and cultured in different conditions and for different times. When present, unless differently indicated, PL was added at a 5% final concentration. Cells were then washed twice with PBS, stained with 50 µL colorant solution (0.75% (g/mL) crystal violet (C3886, Sigma-Aldrich), 0.35% (g/mL) NaCl, 32.3% (*v/v*) absolute ethanol, 8.64% (*v/v*) formaldehyde 37%) for 20 min at room temperature, washed 5 times with water and dried by exposing the plate to air under a chemical hood. To each well 100 µL eluent solution (50% (*v/v*) absolute ethanol and 1% (*v/v*) acetic acid) were added and the absorbance at 595 nm was measured within 10–30 min with a spectrophotometer AD 200 (Beckman Coulter, Brea, CA, USA). Results were expressed as the average of at least three independent experiments performed in quintuplicate on different primary cultures ± SEM values.

#### 4.4.2. Cell Counting

For determination of the osteoblast doubling time in different culture conditions, when the cells reached sub-confluence, they were detached, counted by a hemocytometer and re-plated at a density of 1 × 10^4^ cells/cm^2^. Each experiment was performed in duplicate. Three independent experiments starting from different primary cultures were performed and averages ± standard error of the mean (SEM) calculated. Pictures of cells were taken by a microscope Axiovert 10 (Zeiss, Oberkochen, Germany).

### 4.5. Osteogenic Differentiation

#### 4.5.1. Differentiation In Vitro

Osteoblasts, plated at a density of 5 × 10^4^ cells/well in 24 well plates in the presence of FCS or FCS + PL, and cultured to confluence, were washed with PBS and allowed to differentiate in the absence of PL in osteogenic induction medium (10% FCS, 2 mM L-glutamine, 50 μg/mL penicillin/streptomycin, 50 μg/mL ascorbic acid, 1.5 mg/mL β glycerophosphate, and 10^−7^ M dexamethasone) [58]. After 7, 14, and 21 days of osteogenic differentiation treatment, Alizarin Red staining and quantification were performed following the manufacturer’s instructions.

#### 4.5.2. Differentiation In Vivo

All animal procedures were approved by the IRCCS AOU San Martino-IST “Ethical Committee for animal experimentation (CSEA)” and notified to the Italian Ministry of Health, having regard of the D.lgs 27 January, 1992 n.116/92. The in vivo osteogenic assay was based on an established model of ectopic bone formation [59]. Briefly, we seeded (1.5 or 2.5) × 10^6^ cells onto small porous ceramic cubes (4 × 4 × 4 mm Skelite scaffolds composed of 33% Hydroxyapatite and 67% Silicon stabilized tricalcium phosphate). Cell seeded cubes were implanted subcutaneously on the dorsal surface of immune-compromised mice (CD-1 Nu/Nu, Charles River, Wilmington, MA, USA). The implants were recovered at 60 days. Harvested samples were treated as described by El Backly et al. [60]. Briefly, all samples were fixed in 3.7% paraformaldehyde and dehydrated. Undecalcified samples were embedded with light-curing resin Technovit 7200 VLC (Kulzer, Wehrheim, Germany) and polymerized with EXAKT 520 polymerizator system (EXAKT Technologies, OK, USA). Sections of 30–40 μm thickness were produced using a commercially available cutting grinding system (EXAKT 310 CP cutting and EXAKT 400 CS micro grinding units, EXAKT Technologies). Sections were stained with Stevenel’s blue/Van Geison picrofuchsin stain, staining in red the calcified bone and in blue non calcified bone and osteoid. Images of the sections were acquired using a phase-contrast Axiovert 200M microscope (Zeiss). Scaffolds seeded with cells from two different primary cultures were implanted and analyzed, yielding similar results.

### 4.6. Western Blot Analysis

Osteoblasts were cultured in FCS and at confluence starved in Serum Free medium (SF) for 2 h, before they were incubated for different times in SF medium supplemented with 5% PL. After PL treatment, the cell monolayers were washed twice with cold PBS, and transferred at −80 °C for 10 min. The monolayer cells were then scraped in cold RIPA buffer (50 mM Tris (pH7.5), 150 mM NaCl, 1% Deoxycholic acid, 1% Triton X-100, 0.1% SDS, 0.2% NaN_3_ and proteinase inhibitor cocktail (1:10, P2714, Sigma Aldrich)]. Cells in RIPA buffer were transferred into cold 1.5 mL Eppendorf tubes, lysed for 30 min on ice and centrifuged 10 min at 10,000 rpm at 4 °C. The supernatant containing protein extract was collected and kept at −20 °C until use. Protein concentration was quantified by Bradford assay [61]. For each sample, 30–80 µg of proteins were loaded on the gel. Western blot analysis was performed according to Ulivi et al. [62]. Antibodies dilution adopted were: α-Cyclin D1 (1:250), α- Actin (1:200), HIF-1α (1:500), phospho-STAT3 (1:1,000), STAT3(1:1,000), HRP conjugated anti mouse and anti-rabbit antibodies (both 1:5,000), and VEGF (1:200). Peroxidase conjugated anti-goat antibody (1:10,000).

### 4.7. Nuclear Protein Extraction

Confluent cells were washed with sterile PBS and transferred in SF. Nuclear extract and cytoplasmic fraction were prepared according to the protocol suggested by Active Motif (Carlsbad, San Diego, CA, USA) with little modifications. Briefly, after washing with 10 mL cold PBS, cells were scraped in 10 mL cold PBS, transferred into 15 mL tubes and centrifuged at 300 g for 5 min at 4 °C. The obtained pellet was suspended by gently pipetting in 100 µL ice-cold hypotonic buffer containing 20 mM Tris HCl (pH 7.5), 2 mM ethylene glycol tetraacetic acid (EGTA), 2 mM EDTA, 250 mM sucrose and proteinase inhibitor cocktail (P2714, Sigma Aldrich). The cells were transferred into pre-chilled 1.5 mL Eppendorf tubes and allowed to swell for 15 min. Then 5 µL of 10% Nonidet-P40 were added and the tubes vortexed for 10 s. The cytoplasmic fraction was collected after brief centrifugation (13,000 g) for 30 s at 4 °C and stored at −80 °C. The nuclear pellet was suspended in 20 µL Complete lysis buffer (Active Motif) and lysis performed by rocking the tubes on ice on a lab shaker for 30 min. The tubes were then centrifuged for 10 min at 14,000 g at 4 °C and the supernatant containing the nuclear extract was collected and stored at −80 °C. Concentration of nuclear proteins was measured by the Bradford assay [61].

### 4.8. HIF-1α Binding to DNA

Under hypoxic condition, HIF-1α is protected from hydroxylase degradation, relocates to nucleus and combines with HIF-1β to form an active HIF-1 complex binding to HRE sequences of target genes [30]. We used the TransAM HIF-1 kit to determine the active HIF-1 complex binding to the DNA in the cells treated with PL. 20 µg of nuclear proteins from each sample were added to the wells of a 96-well plate on which an oligonucleotide containing the HRE sequence (5′-TACGTGCT-3′) from the erythropoietin (EPO) gene was immobilized. The HIF-1 binding to the specific sequence was detected by HIF-1α primary antibodies and subsequent incubation with HRP-conjugated secondary antibodies. The absorbance was read with a spectrophotometer AD 200 (Beckman Coulter, Brea, CA, USA) at 450 nm with reference wavelength of 655 nm.

### 4.9. VEGF Secretion

Two parallel and identical confluent cell cultures were washed with PBS and starved in SF medium for 2 h. After the starvation, one culture was maintained in SF medium (Control) whereas the other culture was transferred in SF medium supplemented with 5% PL. After 24 h, both cultures were washed with PBS and further incubated with SF + 0.1% FCS for 16 h. Supernatants were collected in cold 1.5 mL Eppendorf tubes, centrifuged at 4 °C and kept at −20 °C. VEGF detection was performed with the Human VEFG Quantikine ELISA kit by R&D System (Minneapolis, MN, USA) according to the manufacturer instructions. Results are from two different experiments on two different primary cultures. The analysis was performed in duplicate testing two medium dilutions.

### 4.10. Statistics

All data are presented as means and standard error of the mean (SEM). Level of significance was set at *p* < 0.05 (* *p* < 0.05, ** *p* < 0.01, *** *p* < 0.001, and **** *p* < 0.0001). Statistical analysis was performed according to unpaired t-test using online application of the Graphpad software (GraphPad Software, San Diego, CA, USA, www.graphpad.com).

## 5. Conclusions and Perspectives

The combined effect of the PL stimulation on osteoblasts, i.e., resumption of proliferation by quiescent osteoblasts and, in the same cells, activation of angiogenesis inductive pathways and secretion of factors promoting angiogenesis together with osteoblast differentiation and bone formation, is highly suggestive of a similar role played by the platelet released factors in bone repair and provides a rationale to the application of PL as a therapeutic agent in bones damaged as a result of a fracture or a trauma. Indeed, this natural product contains a multitude of factors controlling and regulating basic mechanisms of cell proliferation and differentiation pathways that are reactivated during wound healing. We believe that platelet derived products are potent drugs that must be used to promote regeneration of damaged bone and other tissues. The knowledge of the surprisingly numerous basic cellular events induced by platelet products in different tissues is fundamental for their adoption in clinical practice.

## Figures and Tables

**Figure 1 ijms-21-05123-f001:**
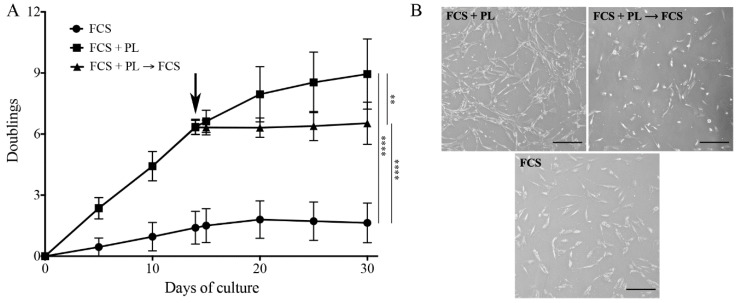
Platelet lysate (PL) induces proliferation of “aged” osteoblasts. (**A**) Growth kinetics plotted as number of cell duplications versus time of culture. At T0, a culture of primary osteoblasts maintained in standard medium in the presence of fetal calf serum (FCS) for 5 passages was split in two and the cultures continued in the presence and in the absence of PL. At T14 (arrow) the PL culture was again split in two and the cultures continued in the presence and in the absence of PL. At each culture time, cells were detached and counted by a hemocytometer. Three independent experiments on 3 different primary cultures were performed. At each time, cell counting was done in duplicate dishes. The average values ± standard error of the mean (SEM) are shown. The significant differences in proliferation rate are also shown, ** *p* < 0.01 and **** *p* < 0.0001. (**B**) Representative images of osteoblasts in the different culture conditions. Scale bar = 50 µm.

**Figure 2 ijms-21-05123-f002:**
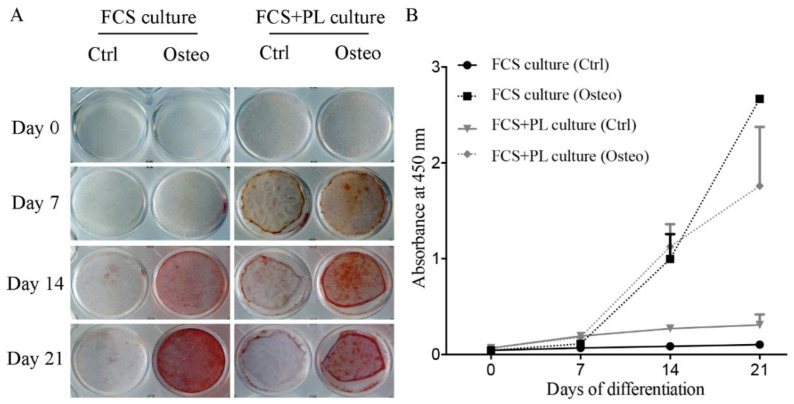
Osteogenic differentiation potential of PL treated and untreated osteoblasts in vitro. FCS culture indicates cells expanded in standard culture medium. FCS + PL culture indicates cells expanded standard culture medium supplemented with PL. Cells from both culture conditions were transferred in standard osteogenic medium. (**A**) Alizarin Red staining, at weekly time intervals, for the two experimental culture groups transferred in osteogenic medium (osteo) or in standard culture medium (Ctrl). (**B**) Quantification of the Alizarin staining. The amount of staining present in each well was determined.

**Figure 3 ijms-21-05123-f003:**
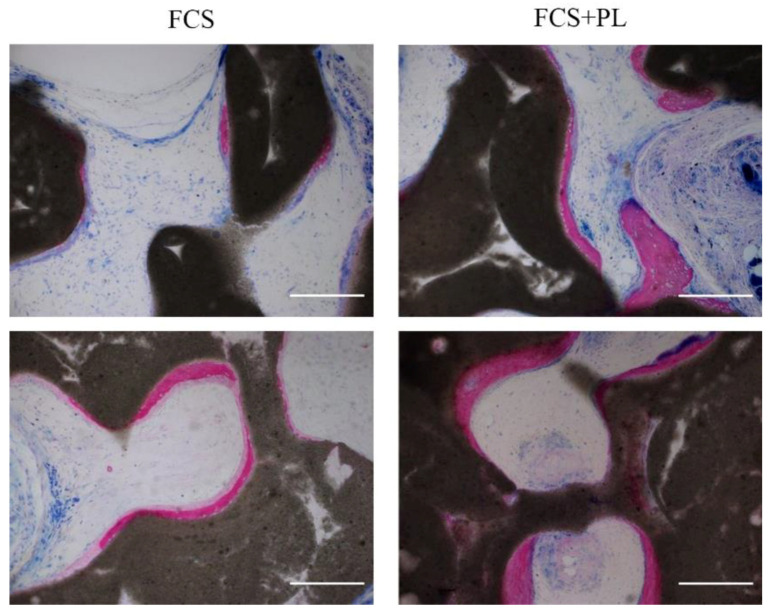
Osteogenic differentiation potential of PL treated and untreated osteoblasts in vivo. Histological analysis by Stevenel’s/Van Gieson staining of ectopic tissue formed after subcutaneous implantation in nude mice of osteoblasts expanded in standard culture medium (left panels) or osteoblasts expanded standard culture medium supplemented with PL (right panels) seeded on osteoinductive scaffolds. 1.5 × 10^6^/scaffold (Upper panels) or 2.5 × 10^6^ cells/scaffolds (lower panels) were implanted. The purple stain refers to the newly deposited calcified bone and the pale pink the non, or only poorly, calcified osteoid (still immature bone). In blue non bone tissues. Scale bar = 200 µm.

**Figure 4 ijms-21-05123-f004:**
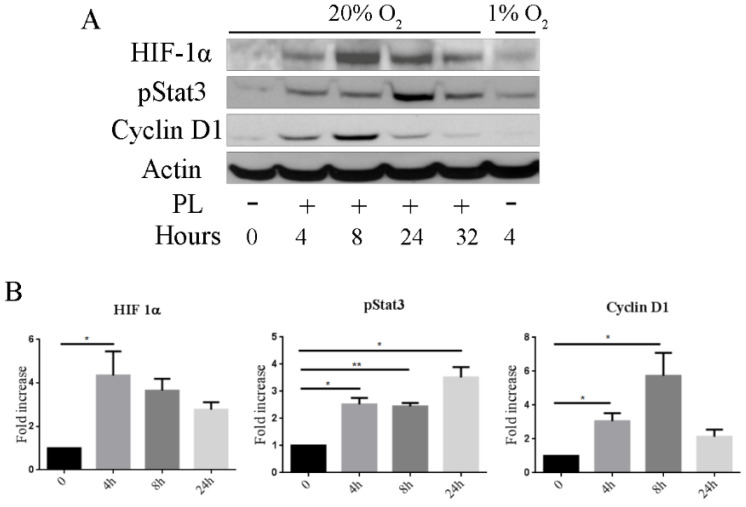
PL induces the activation of hypoxia-inducible factor 1-alpha (HIF-1α) and activation of signal transducer and activator of transcription 3 (STAT3) pathways. (**A**) Western blot analysis of proteins extracted from cells cultured in the absence of FCS and presence of PL for different times. Proteins extracted from cells cultured in hypoxia (1% O_2_) in serum-free medium at 4 h were analyzed as control. (**B**) Quantification of HIF-1α, phosphorylated STAT3 (pSTAT3), and Cyclin D1. Density values are normalized to actin and expressed as fold increase (*n* =3), at 32 h, only two samples were analyzed. * *p* < 0.05, ** *p* < 0.01.

**Figure 5 ijms-21-05123-f005:**
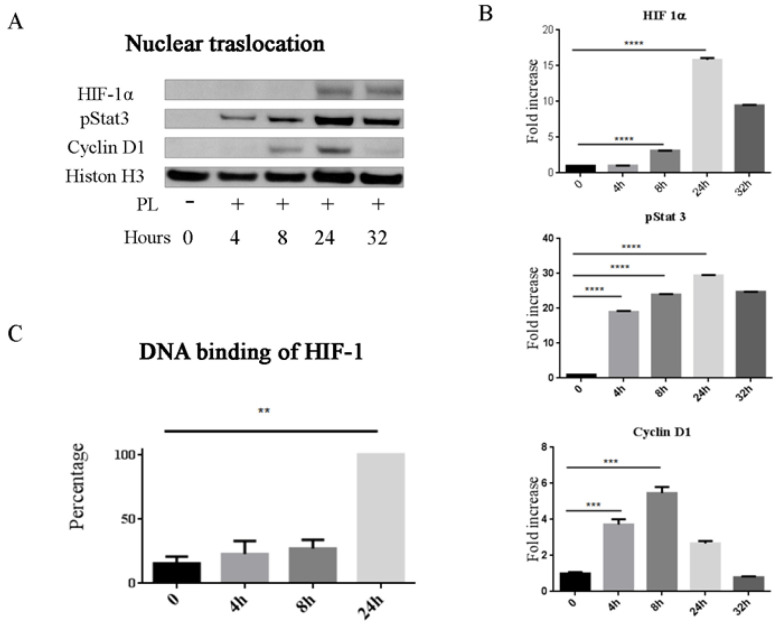
PL promotes stabilization, activation and nuclear translocation of HIF-1α and pStat3. (**A**) Western blot analysis of nuclear proteins extracted from PL treated and control untreated cells. Cyclin D1 and Histone H3 were analyzed in parallel as controls. (**B**) Quantification of HIF-1α, pSTAT3, and Cyclin D1 present in cell nuclei. Density values are referred to histone H3 and shown as fold increase (*n* = 3); at 32 h, only two samples were analyzed. (**C**) Binding to hypoxia-response element (HRE) sequence of the active HIF-1α complexes in nuclear extracts. Data are shown as the percentage of average Optical Density (OD) value at 450 nm of the cells at different times with respect to treated cells at 24h (100%). Two independent experiments were performed in duplicate starting from two different primary cultures. ** *p* < 0.01, *** *p* < 0.001, and **** *p* < 0.0001.

**Figure 6 ijms-21-05123-f006:**
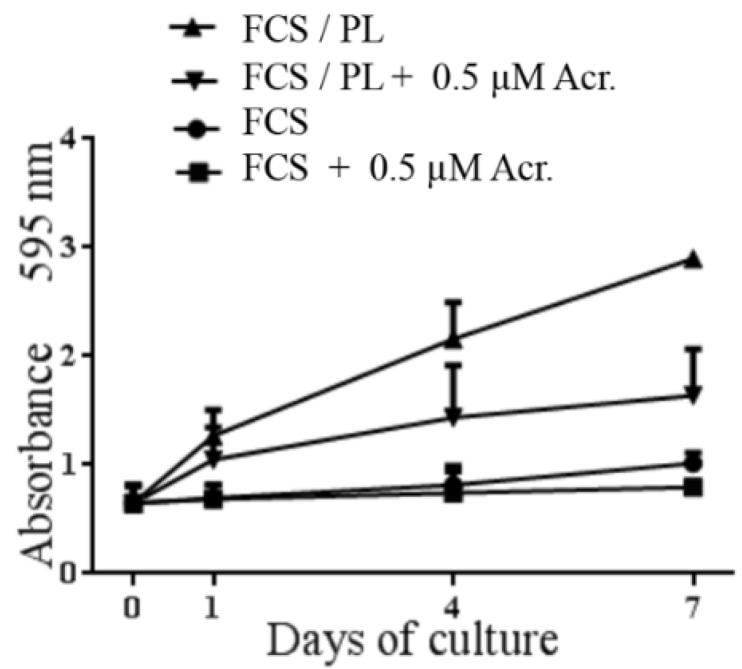
HIF-1α inhibitor acriflavine (Acr.) inhibits osteoblasts proliferation. A Proliferation of the cells in different conditions was determined by Crystal violet assay. Data are the average of three experiments performed in quintuplicate from three different primary cultures.

**Figure 7 ijms-21-05123-f007:**
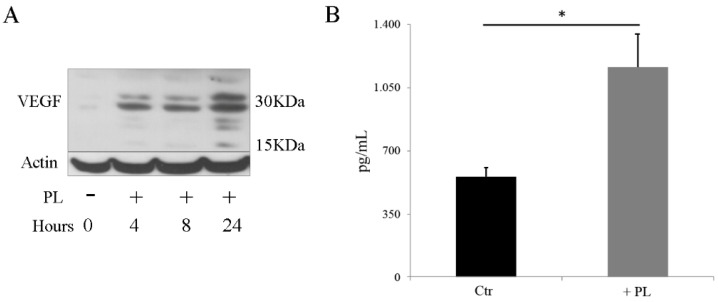
PL induces the expression of vascular endothelial growth factor (VEGF). (**A**) Western blot analysis of proteins extracted from cells expanded in medium containing FCS and transferred in SF medium and in the presence of PL for different times. (**B**) Quantification of VEGF secreted in the cell culture medium. After cell expansion in medium containing FCS, the control culture (Ctr) was transferred and maintained in SF medium whereas the other culture was transferred in SF medium for 2 h and then moved to SF medium supplemented with 5% PL. After 24 h, both cultures were washed with PBS and further incubated with SF + 0.1% FCS for 16 h. Supernatants were collected and VEGF concentration was determined. Results are expressed as pg/mL of supernatant. Two independent experiments were performed on two different primary cultures. Determinations were performed in duplicate at two different dilutions for each experiment. * *p* < 0.05.

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
