# Peer review of "Platelet Lysate Induces in Human Osteoblasts Resumption of Cell Proliferation and Activation of Pathways Relevant for Revascularization and Regeneration of Damaged Bone"

_ijms, 2020, doi:10.3390/ijms21145123_

Round 1
Reviewer 1 Report
The study be Nguyen investigates the effect of platelet-derived molecules on human primary osteoblast functions, including proliferation, differentiation and angiogenesis-related pathways. Osteoblasts were isolated from bone chips from elderly donors (41-85 years of age) undergoing hip joint replacement surgery.
1.The age range of the donors is rather large. How many different donors were used for this study? Were different donor cell populations mixed for one experiment? What was the variation between donors? The authors should include a table indicated which donor was used for which experiment.
2.The in vivo differentiation study was carried out with platelet lysate stimulation, what was the timeframe? What are the different colours in the histological analysis in figure 3? The differences between the histology images are not clear, could the authors perform a quantitative image analysis? How many animals per group were used? Correct spelling mistakes, for example 1.5x106 cells and 2.5x106 cells.
3.The quantification for the 32 hrs timepoint in figure 4b is missing.
4.The authors should include the entire Western blot images for figures 4, 5 and 7 as supplementary items.
5.The quantification for the 32 hrs timepoint in figure 5c is missing.
6.Angiogenesis or bone healing assays should be carried out to confirm the involvement of platelet-derived molecules in osteoblast/endothelial cell interactions.
Author Response
Dear Reviewer,
we addressed your comments into the text hoping to have improved the quality of our manuscript.
Please see the attachment
Best regards
M. Mastrogiacomo

Reviewer 2 Report
This is a well written paper that demonstrated the effect of PL in osteoblast growth and related angiogenic events. Related signaling pathways are well delineated with detailed molecular studies. Several minor comments are as follow.
Line 49-51 : please re-write for clarification
Line 69-74 : please split the sentence and re-write for better understanding
Line 81 : please split the sentence
Line 116 : how can one be sure these cells are osteoblasts ? Tests on specific marker binding seems required.
Line 151 : what is the concentration of PL in the medium ?
Line 267 – 270 : please re-write
Line 336 : Figure 6B : no explanation can be found in the text. What is the intention of showing the stained cells ?
Line 395 – 396 : the thought of this sentence is very unclear
Author Response

(The authors gave the same response as above.)

Round 2
Reviewer 1 Report
The authors have addressed the comments made. The manuscript text requires minor spell checking (e.g. line 162: 1,5x106; line 167: 3,7%; line 290: 1,5x106 and 2,5x106). The number of animals per group should be included in the methods section 2.5.2.